# Phylogenetics and Population Genetics of the *Petrolisthes lamarckii–P. haswelli* Complex in China: Old Lineage and New Species

**DOI:** 10.3390/ijms242115843

**Published:** 2023-10-31

**Authors:** Xuefeng Fang, Dong Dong, Mei Yang, Xinzheng Li

**Affiliations:** 1Department of Marine Organism Taxonomy & Phylogeny, Qingdao Key Laboratory of Marine Biodiversity and Conservation, Institute of Oceanology, Chinese Academy of Sciences, Qingdao 266071, China; fangxuefeng@qdio.ac.cn (X.F.); yangmei@qdio.ac.cn (M.Y.); 2University of Chinese Academy of Sciences, Beijing 100049, China; 3Laoshan Laboratory, Qingdao 266237, China

**Keywords:** *Petrolisthes*, cryptic species, genetic diversity, gene flow, divergence time

## Abstract

*Petrolisthes lamarckii* (Leach, 1821) and *P. haswelli* Miers, 1884 are a pair of sister species of porcelain crabs, both of which are common in the intertidal zone of southern China, typically found under rocks and in the crevices of coral reefs. However, the distribution, genetic relationship and diversity of the two species in China have not been rigorously studied. Meanwhile, *P. lamarckii* is considered as a complex of cryptic species due to their diverse morphological features. In this study, we identified 127 specimens of the *P. lamarckii*–*P. haswelli* complex (LH complex) and recognised a new species through morphological and molecular analysis. Furthermore, we constructed a time-calibrated phylogeny of the LH complex using three mitochondrial and two nuclear genes from all three species, finding that the divergence of the LH complex can be traced back to the Miocene epoch, and that the genetic diversity increased during the Mid-Pleistocene transition period. Glacial refugia formed during the Pleistocene climatic oscillations has been regarded as one of the contributing factors to the diversification of marine organisms in the north-western Pacific. *Petrolisthes haswelli* demonstrates a wide distribution along the southern coast of China, while other lineages display more restricted distributions. The research on the demographic history and gene flow of *P. haswelli* revealed that the Chinese coastal populations experienced an expansion event approximately 12.5 thousand years ago (Kya) and the asymmetrical gene flows were observed between the two sides of the Taiwan Strait and Qiongzhou Strait, respectively, which is likely influenced by the restriction of ocean currents.

## 1. Introduction

The northwest Pacific marginal seas account for about 75% of the global marginal seas [1]. The marginal seas of China are among the most extensive in the northwest Pacific. Their oceanography and remarkable geological history are highly distinctive. Therefore, these marginal seas provide a model environment for studying the phylogeography and genetic diversity of marine organism populations [2]. Recent efforts have been made to study, discover, and describe cryptic species. Cryptic diversity is unevenly distributed among organisms, especially in some groups such as amphibians, reptiles, and crustaceans [3]. Indeed, phylogeographical studies have revealed cryptic species diversity within various taxa in crustacea [4,5]. Phylogeographical analyses incorporating geographically dense and large sample sizes, alongside molecular genetics-based phylogenies, have made substantial contributions to our understanding of species history and speciation [6,7]. Constructing time-calibrated phylogenies can provide valuable insights into the divergence of lineages both between and within oceanic areas [8]. Research on genetic diversity can also reveal species’ distribution patterns and the status of gene flow, helping to understand the impact of past and present environments on population differentiation [9,10,11]. In addition, the haplotype network and principal component analysis (PCA) can provide more intuitive insights into the differentiation among different populations of a species [12,13]. Overall, diversification in marine ecosystems may occur along both geographic and ecological partitions [14]. However, the combination of geographic and ecological partitioning is probably perceived as the most impactful factor driving speciation or intraspecific sub-division [15,16]. Numerous questions regarding the population structure and connectivity within marine species remain unresolved [17]. Repeating drastic changes and complex ocean currents together shape the abundant geographic variation and ecological adaptation of marine animals. The Pleistocene climate fluctuations have caused significant changes in the coastlines and configurations of these marginal seas [18,19]. In addition, the direction of ocean currents also has some influence on gene flow between populations, shaping the current biogeographic patterns of species and genetic biodiversity [11,16]. The ocean current in the Qiongzhou Strait flows predominately from east to west all year round [20]. When the northeast winter monsoon winds prevail, the cold China Coastal Current is driven to flow southward into the South China Sea through the Taiwan Strait. When the southwest summer monsoon winds prevail, the southern part of the China Coastal Current is reversed and flows northward into the East China Sea [21].

Porcelain crabs exhibit a worldwide distribution, occupying both tropical and temperate marine environments and represent one of the most diverse families of decapod crustaceans in near-shore areas. The genus *Petrolisthes* is characterised by high biodiversity, with over 100 recognised species distributed throughout the world in both tropical and temperate sea areas [22]. *Petrolisthes lamarckii* and *P. haswelli* are a pair of sister species, both of which are common in the intertidal zone of southern China, typically found under rocks and in the crevices of coral reefs. *Petrolishtes haswelli* is widely distributed throughout the tropical and subtropical regions of the West Pacific, while *P. lamarckii* is an Indo-Pacific species, ranging from the Red Sea to the Tuamotu Islands [23,24], although some distribution records need to be verified. These two species are morphologically very similar, and are sometimes difficult to distinguish from each other when preserved in ethanol [25]. Osawa and Chan (2010) compared *P. lamarckii* and *P. haswelli* and concluded that *P. haswelli* was more strongly convex, striate, and densely setose on the dorsal surface, and the rostrum was more depressed ventrally [25]. On the other hand, the taxonomy of *P. lamarckii* has long been considered unsatisfactory [25]. Borradaile (1898) once established three subspecies under *P. lamarckii*, forming a group of species or variations called the *P. lamarckii* complex [26]. However, those subspecies were morphologically very distinct from *P. lamarckii*, and then Kropp (1984) elevated them to species [27]. Nevertheless, a few more morphological variations were observed and still considered intraspecific. Osawa and Chan (2010) described two morphological forms of *P. lamarckii* based on specimens from the Taiwan Island, indicating the presence of cryptic species [25]. Meanwhile, as geographically wide distributed species, *P. lamarckii* and *P. haswelli* from different localities were recognised only based on morphological characters in previous works (e.g., [25,28]). The genetic connectivities among different geographic populations are still unknown, and the genetic consistency for each species has not been verified. Similarly, due to the scarcity of molecular data, the phylogenetic relationship within this species group is also poorly understood. 

Here, we treated multiple populations of *P. lamarckii* and *P. haswelli* together as a single species complex (LH complex). We used integrative methods combining the taxonomic, phylogenetic, and population genetic analysis to explore the biodiversity and distribution pattern of LH complex in China; to investigate the divergence time and gene flow among geographic populations; finally, to address the taxonomic issues of this complex and describe a cryptic new species.

## 2. Results

### 2.1. Genetic Diversity

We sequenced 135 samples of four *Petrolisthes* species (*Petrolisthes haswelli*, *P. lamarckii*, *P. polychaetus* and *P. shanyingi* sp. nov.) collected along the coast of the southern Chinese mainland and Hainan Island from the rocky intertidal and from coral reefs (Figure 1), and obtained new sequences from 132 individuals at the *COI* locus, 105 individuals at the *16S* locus, 98 individuals at the *Cytb* locus, 57 individuals at the *18S* locus, and 91 individuals at the *H3* locus. In total, 28 unique *COI* haplotypes, 18 unique *16S* haplotypes, 29 unique *Cytb* haplotypes, 1 unique *18S* haplotypes, and 4 unique *H3* haplotypes were detected. None of the *COI* sequences exhibited the characteristics of nuclear pseudogenes (frame shifts or premature stop codons). The haplotype diversity and nucleotide diversity were presented in Table 1. In addition, we compared the nucleotide diversity of different gene markers. The results showed that *Cytb* (0.0792) had the highest value, while *COI* (0.0589) and *16S* (0.0384) were similar. *H3* (0.0053) and *18S* (0), on the other hand, had very low values.

The range of pairwise F_ST_ values between area-populations of *Petrolisthes haswelli* varied from 0.000 to 0.482. However, only the result between BH and HN is significant. The genetic distance ranged from 0.000 to 0.001 between area-populations (Table 2). The genetic distance and pairwise F_ST_ values based on the *COI* sequence indicate that there is no significant genetic differentiation among area-populations. Higher values of F_ST_ were observed between HN and other area-populations, indicating the presence of genetic divergency between them.

According to the phylogenetic results (see below), we calculated the genetic distance among different genetic lineages based on *COI* sequences. The genetic distance of the four species ranged from 0.132 to 0.161 (see the section on systematics). Within the species, the genetic distances were as follows: 0.000–0.004 (*P. haswelli*), 0.002–0.011 (*P. lamarckii*), 0.000–0.009 (*P. polychaetus*) and 0.000–0.019 (*P. shanyingi* sp. nov.).

### 2.2. Phylogeny, Species Delimitation and Divergence Time Estimation

Phylogenetic trees (Figure 2) produced by maximum likelihood (ML) analyses and Bayesian Inference (BI) based on all dataset were all virtually congruent, with minor differences mainly in terminal intraspecific clades. In addition, we, respectively, constructed phylogenetic trees with a single gene and Mito dataset by ML methods (Appendix A). In the phylogenetic tree, each of the four species was clearly separated, while the *P. lamarckii* clade and *P. haswelli* clade were clustered together, forming a high support clade (pp = 1; bp = 96%). Meanwhile, *P. shanyingi* sp. nov. was sister to the *P. lamarckii*–*P. haswelli* clade with high support (pp = 1; bp = 90%), indicating a closer relationship within the LH group (Figure 2).

Two independent species-delimitation methods (GMYC and bPTP) based on the dataset showed similar results and recovered four or five *Petrolisthes* molecular lineages of our samples (Figure 2). As for the lineage *P. shanyingi* sp. nov., both the bPTP with the ML tree and BI tree analyses considered the clade as a single species. In contrast, the GMYC analyses split the clade into two species. The PCA analysis also divided all samples into four groups, which is consistent with the phylogenetic results (Figure 2 and Appendix A). These are all congruent with the morphological distinctions among those species, suggesting the validity of the interspecific characteristics and the establishment of the new species (see Section 3).

Based on all the dataset, the most recent common ancestor (tMRCA) of the LH complex was estimated to be around 17.0 Mya (95%HPD: 8.1–29.2 Mya). *P. haswelli* and *P. lamarckii* diverged approximately 12.0 Mya (95%HPD: 5.8–21.4 Mya), and their common ancestor separated with *P. shanyingi* sp. nov. around 17.0 Mya (95%HPD: 8.1–29.2 Mya). Furthermore, the divergence times within species (0.5–1.5 Mya) are concentrated in the Pleistocene epoch (Figure 3).

The following mean substitution rates were estimated for each gene: 6.2 × 10^−3^ (*COI*), 1.9 × 10^−3^ (*16S*), 7.4 × 10^−3^ (*Cytb*), 4.2 × 10^−4^ (*18S*), and 4.8 × 10^−4^ (*H3*).

### 2.3. Geographical Distribution of Genetic Biodiversity

The TCS network exhibited four different haplotype groups corresponding to four *Petrolisthes* species (*P. haswelli*, *P. lamarckii*, *P. polychaetus*, and *P. shanyingi* sp. nov.), which is congruent with the phylogenetic and PCA analyses. Among all the specimens we obtained from the southern coastal areas of China, *P. haswelli* has the widest distribution range. It is noteworthy that the Australian haplotype was shared by two local populations in China and shared a more distant common ancestor with other branches in the *P. haswelli* clade (Figure 4). On the other hand, one dominant haplotype was observed covering all the sampling regions. However, fewer haplotypes were observed in *P. haswelli* under such a wide sampling range and large sample size compared to the other three species. The other three species have a relatively limited distribution range in China, which are geographically restricted to the south of the Taiwan Strait but possess higher haplotype diversities; nearly every individual represents a unique haplotype, especially for those populations in Hainan Island. 

### 2.4. Demographic History and Gene Flow in P. haswelli

The estimated Tajima’s D is −1.41 (*p* > 0.1) and Fu’s Fs is −2.52 (*p* > 0.1). They were consistent, but both were not significant. The Bayesian Skyride Plot for the analysed populations indicated that *P.haswelli* populations remained relatively stable with a slight increasing trend before 12.5 Kya, while a population expansion event occurred between 5–12.5 Kya (Figure 5).

For the gene flow analyses in Migrate-n, the results revealed asymmetric gene flow among the four area-populations (Figure 6). Significant migration was observed from the pairwise area-populations. The Nm from DS to the adjacent area-population XG (20.24) is more than seven times the one from XG to DS (2.60). The Nm between DS and EC (21.13, 16.15), as well as EC and HN (16.58, 24.41), were similar. The Nm from EC to the adjacent area-population BH (14.65) is more than thirty-nine times the one from BH to EC (0.37). More significantly, the Nm from HN to the adjacent area-population BH (23.11) is more than sixty-seven times the one from BH to HN (0.34). Effective population size for each group revealed the smallest in the XG population (*θ* = 0.00130), relative to the values of the DS, EC, BH and HN populations, respectively (0.02419, 0.03658, 0.01058 and 0.05124, Figure 6).

## 3. Systematics

Porcellanidae Haworth, 1825 [30]

*Petrolisthes* Stimpson, 1858 [31]

### 3.1. Petrolisthes lamarckii (Leach, 1821)

Figure 7A and Figure 8C.

#### 3.1.1. Synonymy

*Pisidia lamarckii* Leach, 1821: 54 (type locality: New Holland (western and northern coast of Australia)) [32].

*Porcellana dentata* H. Milne Edwards, 1837: 251 (type locality: Java) [33].

*Porcellana pulchripes* White, 1847: 129 (nomen nudum) [34].

*Porcellana speciosa* Dana, 1852: 417 (type localities: Drummond Island, Kingsmills Group; Wake Island; Mangsi Islands, Philippines; Raraka, Tuamotu Archipelago) [35]; 1855: pl. 26, figure 8 [36].

*Porcellana bellis* Heller, 1865: 76 (type locality: Nicobar) [37].

*Petrolisthes lamarckii*—Miyake 1942: 342, text—figures 7 and 8 [38]; 1943: 98, figure 29 [39].—Kropp 1984: 100 [27].—Haig 1992: 315, figure 11 [28].—Hsieh et al. 1997: 326, figures 24F and 28 [40].—Osawa and Chan 2010: 142 (part), figures 109–111 and 113d–f [25].—Beleem et al. 2016: 5, figure 2d [41].

#### 3.1.2. Material Examined

Holotype, sex indetermined, (CL 7.2 mm), western and northern coast of Australia (no exact location), collection date unknown, from Leach’s (1821) collection.

JJD-HS, two males (CL 6.6–7.3 mm), Jiajing Island, Wanning, Hainan Province, in the crevice of coral reefs, within 10 m, 28 March 2018. LHT-HS, two males (CL 4.8–6mm), one ovig. female (CL 10.7 mm), Luhuitou, Sanya, Hainan Province, under stone, 30 March 2018.

#### 3.1.3. Diagnose

Carapace ovate, as long as or slightly longer than broad; lateral branchial margins slightly convex; dorsal surface covered with numerous transverse striae usually bearing short setae; pair of epibranchial spines well developed at anterior ends of cervical grooves. Rostrum sinuously triangular; apex strongly bend ventrally. Chelipeds subequal and hairless, with numerous weak, short rugae on dorsal surface; merus with low, distal lobe on dorsoflexor margin; carpus armed with 3–5 (usually 4) teeth on dorsoflexor margin and three spines on distal half of dorso-extensor margin; palm broad, hairless, unarmed on extensor margin. Ambulatory legs moderately slender; meri unarmed on extensor margin, lateral flexor margin with acute distal spine on first and second legs, but unarmed on third leg; carpi armed with distal spine on extensor margin of only first leg; propodi with four movable corneous spines including distal pair on flexor margin; dactyli with three corneous spines on flexor margin.

Coloration: Dorsal surface of carapace and pereopods with scattered black or brown spots. Background colour varying (light brown based on our specimens), generally forming a fine reticulated colour pattern. Palm with row of orange spots along dorso-extensor margin. Propodi of ambulatory legs with transverse, purple colour bands.

Habitat: Intertidal and subtidal zones, under rocks and in crevices of coral reefs; depth within 10 m (present study).

#### 3.1.4. Distribution

Widely distributed in the tropical Indo-West Pacific, extending from the western coast of Africa to the Line and Tuamotu Islands.

#### 3.1.5. Remarks

Osawa and Chan (2010) noted there existed two morphological forms of *P. lamarckii* from specimens of Taiwan Island [25]. The first form has three broad teeth on the dorsoflexor margin of the cheliped carpus, blunt and a small distal spine on the lateral flexor margin of the merus of the first ambulatory leg, and the carpus of the first ambulatory leg is unarmed. The second form has four or five broad teeth on the dorsoflexor margin of the cheliped carpus, an acute distal spine on the lateral flexor margin of the merus of the first ambulatory leg, and an acute disto-extensor spine on the carpus of the first ambulatory leg. Furthermore, the second form has a row of orange spots along the dorso-extensor margin of the chela. Our *P. lamarckii* specimens agree well with the second morphological form. More detailed differences are discussed under the remarks of *Petrolisthes shanyingi* sp. nov. (see below).

Leach (1821) only gave a brief description of *P. lamarckii*, making it difficult to find out which morphological form belongs to the real *P. lamarckii* [32]. White (1847) listed out *Porcellana lamarckii* (=*Pisidia lamarckii* Leach, 1821) which was deposited in the Natural History Museum (NHM) London, which included the type material from Leach’s collection [34]. We re-examined the type material of the *P. lamarckii* from the NHM (catalogue No. White 1 112; pers. comm. with Paul Clark), and ascertained that the type locality was New Holland, a place where it was first applied to the western and northern coast of Australia. This specimen was taken as the holotype in NHM and no other type material was found. The holotype, however, is now in such poor condition that only the carapace (severely broken) and a third (probably) ambulatory leg are left. The morphological characters that can be confirmed are: somewhat subtriangular rostrum, no spines on the merus and carpus of that ambulatory leg, and the dorsal branchial regions bearing sparse long setae. However, it is still insufficient to define the *P. lamarckii*’s morphological features. Considering that the holotype is no longer morphologically representative, new specimens of *P. lamarckii* should be collected from the type locality and redescribed based on Leach’s (1821) and Kropp’s (1984) descriptions in the future [27,32]. Here, we refer to Kropp’s (1984) species delimitation and description on *P. lamarckii*, who provided a detailed review of the *P. lamarckii* species complex throughout the Indo-West Pacific [27]. According to Kropp’s description, *P. lamarckii* has a distal spine on the carpus of the first ambulatory leg, and the palm of the cheliped has a row of colour spots along the extensor margin. These characters are consistent with those of our *P. lamarckii* specimens and Osawa and Chan’s (2010) second morphological form, although Kropp’s (1984) specimens might be variable on the fresh coloration [25,27]. It is noteworthy that Leach (1821) described three teeth on the dorsoflexor margin of the cheliped carpus [32]. Kropp (1984) and Miyake (1942) recorded three and infrequently four teeth on this margin (Kropp: specimens from Mariana; Miyake: specimens from around Palau), while Osawa and Chan (2010) and Haig (1992) described or illustrated four or five teeth (Haig: specimens from Hong Kong) [25,27,28,38]. Among our *P. lamarckii* specimens, only one cheliped out of five specimens has three teeth, while all others have four teeth, although the distal tooth is reduced.

*Petrolisthes lamarckii* is morphologically similar to *P. haswelli*, and sometimes it is hard to distinguish between them [25]. The two species have a closer genetic relationship than that shared by *P. lamarckii* and *P. shanyingi* sp. nov. (see Result). For fresh specimens, *P. lamarckii* can be easily distinguished by its row of colour spots on the dorso-extensor margin of the palm [25]. Furthermore, *P. haswelli* is more setose on the dorsal surfaces of the carapace, chelipeds and extensor margins of the ambulatory legs, and the palm is more granulate on the dorsal surface and significantly flattened on the lateral part [25]. *Petrolisthes haswelli* also usually has truncate teeth on the dorsoflexor margin of the cheliped carpus, and one or more of those teeth sometimes have an affiliated tooth or distinct notch on the anterior margin. Our specimens of *P. haswelli* exhibit variable colour patterns among different geographic populations (Figure 7B–E), but their genetic distance are not significant (0.000–0.004).

### 3.2. Petrolisthes shanyingi sp. nov.

Figure 7F, Figure 8A,B,E–I and Figure 9.

#### 3.2.1. Synonymy

*Petrolisthes lamarckii*—Yang 1983: 3 (part), pl. 4 [42].—Yang and Sun 2005: 8, fig. 5 [43].—Werding and Hiller 2007: 10, fig. 8 [24].—Osawa and Chan 2010: 142 (part), figs. 112, 113a–c [25].—Prakash et al. 2013: 2, fig. 2B [44].

#### 3.2.2. Material Examined

Holotype. LHT-HS5, one male (CL 6.9 mm), Luhuitou, Hainan Province, under stone, 30 March 2018.

Paratypes. SYHS-HS, two males (CL 5.2–5.6 mm), one ovig. female (CL 5.9 mm), Houhai Village, Sanya, Hainan Province, in the crevices of coral reef, 22 March 2018. LHT-HS, Collected with holotype, 14 males (CL 3.3–8.2 mm), one female (CL 3.7 mm), three ovig. female (CL 4.6–8.7 mm), Luhuitou, Sanya, Hainan Province, under stone, 30 March 2018.

#### 3.2.3. Description

Carapace: generally glabrous, ovate, as long as or slightly longer than broad; lateral branchial margins slightly convex, divergent posteriorly and broadest at posterior branchial regions. Dorsal surface flattish, finely covered with numerous transverse striae, striae relatively short and indistinct around cardiac and intestinal areas; cervical grooves distinct; pair of epibranchial spines well developed; posterior branchial regions obliquely rugose on lateral parts; posterior margin moderately concave. Rostrum subtriangular, approximately one-third carapace width, with round apex, strongly bent ventrally; lateral margins sinuous. Orbits shallow; supra-ocular margins smooth and oblique; outer orbital angles blunt; median frontal groove distinct, extending from the apex of the rostrum and dividing the protogastric ridges into two parts. Pterygostomian flaps are whole, with an elevated ridge along the anterior dorsal margin; dorsal half of surface depressed, ventral half with longitudinal, sinuous ridges. 

Third thoracic sternite: broad, trilobate; median lobe with broad triangular apex, lateral lobes slender, produced slightly beyond median lobe anteriorly.

Telson: composed of seven plates arranged.

Eyes: large, ocular peduncles short, as broad as cornea.

Basal article of antennular peduncle: slightly longer than broad, with transverse and oblique ridges on anterior half of ventral surface; anterior margin weakly serrated; lateral margin medially convex.

Antenna1 peduncle: first article immovable, not projecting; second article (first movable article) with small lamellar lobe on anterior margin; third article with straight anterior and posterior margins.

Third maxilliped: ischium broadly ovate, rugose transversely on ventral surface. Merus to dactylus with long setae on flexor margin. Merus with extensor margin slightly convex medially; flexor margin projecting into triangular lobe; lateral surface with transverse rugae. Carpus with flexor margin convex, bearing small, triangular sub-proximal projection; lateral surface with rugae entirely. Propodus with sparse short rugae on lateral surface. Dactlylus relatively smooth on the lateral surface.

Chelipeds: subequal and hairless. Merus short; dorsoflexor distal lobe low and rounded; ventrodistal margin with the strong median spine; dorsal and ventral surface with transverse rugae. Carpus subrectangular, approximately 1.9 times as long as broad; dorsoflexor margin with 3 broad, tip-rounded teeth along proximal 2/3 length, proximal tooth prominent and triangular, second tooth comparatively low (half height of proximal tooth), distal tooth vestigial or at most subequal to second one; dorso-extensor margin ridged, with oblique elevated rugae on proximal half, and three acute spines on distal half, distalmost spine at end of margin; dorsal and ventral surfaces with numerous transverse or oblique scale-like rugae, rugae on dorsoflexor half relatively short. Palm broad, approximately 1.8 times as long as broad, 1.3 times longer than carpus; extensor margin entire, compressed dorsoventrally; flexor margin moderately inflated dorsoventrally; dorsal surface moderately swollen along midline, dorsal and ventral surfaces with numerous scale-like rugae, rugae on flexor parts of both sides relatively longer. Fingers approximately half of the palm length; tips hooked and crossed to each other distally; dorso-extensor margin of dactylus ridged, obliquely rugose, with a shallow submarginal groove; dorsal and ventral surfaces of both fingers covered with short rugae; occlusal edges crenulate, weakly convex on the fixed finger and bearing triangular basal tooth on dactylus.

Ambulatory legs (P2–4): sparsely setose; surface of each segment covered with numerous, long or short rugae. Meri slender and subrectangular; P2 merus approximately 0.6 times that of the carapace length, P3 merus 1.1 times longer than P2 merus and P4 merus slightly shorter than P2 merus; P2 merus 2.3 times longer than broad (length/breadth ratios: 2.0 on P3 and 1.8 on P4); extensor margin gently curved, unarmed, bearing few plumose setae; flexor margin relatively straight, unarmed or with small distal spine on P2 or P3. Carpi unarmed, approximately half of merus length on each leg, P3 carpus slightly longer than P2 and P4 carpi; lateral surface with longitudinal ridges on the distal part. Propodi stout, approximately 0.7 times of merus length on each leg, P3 propodus 1.1 times longer than P2 and P4 propodi; flexor margin with four movable corneous spines including distal pair; distal end of lateral surface usually bearing four stiff setae. Dactyli each terminating in strong curved claw, 0.4 times propodi length; flexor margin with three corneous spines; lateral surface with row of fine setae.

Pleopods: males with pair of well-developed pleopods modified as gonopods on second abdominal segment; endopod elongate, tapering distally, marginal setose; exopod small, ovate. Females with pairs of slender pleopods on each third to fifth abdominal segments.

Coloration: Brown, reddish brown or rusty red on dorsal surface of whole body. Propodi of ambulatory legs with transverse purple bands.

Habitat: Intertidal zones, under rocks or in crevices of coral reefs.

#### 3.2.4. Distribution

Tropical Indo-West Pacific. Red Sea; Lakshadweep, Indian Ocean; Xisha Islands, Hainan Island and Taiwan island, northern part of the South China Sea.

#### 3.2.5. Etymology

The new species is named after the Shanying College, the Middle School Affiliated to Qingdao University, in appreciation of their support for our marine biodiversity study.

#### 3.2.6. Remarks

*Petrolisthes shanyingi* sp. nov. is morphologically similar to *P. lamarckii* and *P. haswelli* in the following characters: the rostrum is roughly triangular; the carapace and the rostrum are unarmed except the epibranchial spines; the carapace and chelipeds covered with striae or rugae on the dorsal surface; the cheliped carpus has at least three teeth on dorsoflexor margin and three spines on dorso-extensor margin; the cheliped palm is unarmed and hairless on the extensor margin; the meri of ambulatory legs are unarmed on the extensor margins. *Petrolisthes shanyingi* sp. nov. was previously assigned to *P. lamarckii* because it has only three teeth on the dorsoflexor margin of the cheliped carpus (*P. haswelli* has at least four teeth), and the carapace and pereopods are relatively less setose. The new species can be distinguished from the two relatives by their obtuse, round-tipped teeth on the dorsoflexor margin of the cheliped carpus (sometimes the proximal-most one may be acute) and the distal two teeth are low and vestigial; the P2 merus is unarmed or only has small blunt distal spine on the lateral flexor margin; and the P2 carpus is unarmed on the extensor margin. In contrast, *P. lamarckii* and *P. haswelli* have acute triangular teeth on the dorsoflexor margin of the cheliped carpus and the teeth are relatively produced and narrow; the P2 merus bears a more developed distal spine on the lateral flexor margin; the P2 carpus is armed with an acute distal spine on the extensor margin. In addition, *P. lamarckii* and *P. haswelli* usually have black and dark brown spots on the dorsal surfaces of the carapace and chelipeds, while the new species has no such colour spots.

Our new species agrees with the first morphological form of Osawa and Chan (2010)’s specimens, which has three broad, blunt teeth on the dorsoflexor margin of the cheliped carpus and no disto-extensor spines on the P2 carpus [25]. Osawa and Chan (2010) also described that this form has trilobate rostrum and a blunt, small distal spine on the lateral flexor margin of the P2 merus [25]. The rostrum of our new species, however, seems more likely to be in a sinuously triangular shape (Figure 8B), rather than trilobate which is a feature of *P. polychaetus* (Figure 8D). The lateral lobes of the rostrum, however, are somewhat more strongly produced anteriorly than those of *P. lamarckii* (Figure 8C) and *P. haswelli*. On the other hand, the armature on the meri of the P2 and P3 seems variable in the new species. We observed both a rounded corner and small spine on the lateral distoflexor margins of the meri. Roughly one-third of the specimens of the new species have no such spine, whereas the distal spine on the P2 carpus is always absent in the new species.

Werding and Hiller (2007) reported *P. lamarckii* from the Red Sea, but judging from their illustration, the specimen has three broad, low teeth on the dorsoflexor margin of the cheliped carpus, and unarmed P2 carpus [24]. It seems more likely to be our new species. Similarly, Prakash et al. (2013) reported *P. lamarckii* from the Agatti Island, India, showing a similar morphological form with *P. shanyingi* sp. nov, but the coloration pattern of the specimen is different [44]. Both *P. lamarckii* and *P. shanyingi* sp. nov. are probably widely distributed throughout the Indo-Pacific. However, further taxonomic and phylogenetic studies are still needed to confirm the systematic status of each geographic populations, as well as their genetic relationships.

## 4. Discussion 

### 4.1. Genetic Diversity and Population Structure

Higher levels of genetic diversity are of great importance to species survival and reproduction under environmental stress [45]. Coykendall et al. (2017) found that the genetic diversity of haplotypes across three *Munida* species was particularly high in the mitochondrial *COI* region and this is especially noteworthy within the *Munida* genus (the genus *Munida* and *Petrolisthes* both belong to the superfamily Galatheoidea), where a high level of polymorphisms in *COI* may be attributed to a fast mutation rate, large population sizes, or an ancient origin that allows for the accumulation of mutations over time [46]. Our study also revealed apparent variation in the *COI* region among different *Petrolisthes* species, as well as in other mitochondrial genes such as *16S* and *Cytb*. There are differences in the nuclear genes/markers between species, while the intraspecific difference is almost inexistent. Therefore, although the *18S* and *H3* gene can be used to distinguish between different species, they may not be as effective in studies of genetic diversity at the intraspecific level. Haplotype and nucleotide diversities constitute a crucial aspect of genetic diversity, believed to be indicators of population size, history, ecology, and the capacity of populations to adapt to environmental changes [47]. The level of genetic diversity is influenced by numerous factors, including environmental conditions, genetic drift within populations, bottleneck effects, inbreeding, and life history traits. Furthermore, due to morphological similarities, the species diversity has been severely underestimated, with some crustaceans previously thought to be widespread actually comprising a complex of cryptic species [48,49]. In this study, we involved *Petrolisthes haswelli*, *P. lamarckii*, *P. polychaetus* and *P. shanyingi* sp. nov. populations in the PCA and network analysis and recovered distinct four groups, indicating the genetic divergence of those related species. The result was congruent with that of the phylogenetic analysis. Another study on *Petrolisthes* species also indicated that, while the 12–19 day planktonic larval stage of *Petrolisthes armatus* promotes a high population structure, it alone cannot fully explain the significant divisions observed in this species [13,50]. However, the populations of LH complex from Hainan Island exhibited relatively high levels of genetic diversity. 

### 4.2. Phylogeny and Divergence Time Estimation

The two species delimitation models (GMYC and bPTP) have been proved effective and have been frequently applied in the species boundary analysis of munidid and munidopsid squat lobsters, which are sibling families of Porcellanidae [51,52]. The species boundary analysis can provide a statistic advantage in addition to the cladistic method and comparison of pairwise genetic distance. The LH complex shares similar habitats and some key morphological characters, making it difficult to distinguish between them, especially when preserved in ethanol, and *P. shanyingi* sp. nov. and *P. lamarckii* were previously assigned to be the same species [25]. In this study, both delimitation models were clearly divided those into three species: *P. haswelli*, *P. lamarckii*, and *P. shanyingi* sp. nov. The GMYC analysis further divided *P. shanyingi* sp. nov. into two clades, but considering the morphological similarity and low genetic distance (0.015), we prefer treating this lineage as one species. Interestingly, *P. lamarckii* was not clustered together with the new species in the phylogenetic tree, but formed a monophyletic clade with *P. haswell*, showing that these two species are more closely related. This indicates that some morphological similarities are not always reliable to estimate the species’ relationship within this complex, for example, the number of teeth on the dorsoflexor margin of the cheliped carpus. 

Conclusions on the timing of speciation events should be approached with caution, as there are certain limitations to our findings. Our relaxed molecular clock method was based on a single calibration, which was the geminate populations of *P. armatus*. Additionally, we utilised broad priors for its date, resulting in wide highest posterior density intervals for our estimates of the most recent ancestor’s time. The analysis suggests that the divergence time of the four *Petrolisthes* species can be traced back to the early Miocene. The MRCA of the LH group diverged during the middle Miocene and the speciation of *P. lamarckii* and *P. haswelli* started in the late Miocene. This result is consistent with previous conclusions concerning the high species diversity of Indo-West Pacific fauna, at least for molluscs, as was well established by the Late Miocene [53,54,55]. The deep-sea squat lobsters were also observed undergoing a remarkable lineage cladogenetic event during the Miocene in the West Pacific [56]. Intraspecific phylogeographical studies have been conducted to investigate biogeographical hypotheses and assess the influence of Earth-history events on the distribution of genetic variation within species [57,58]. It is also noteworthy that the four *Petrolisthes* species underwent an increase in haplotype or genetic diversity within the time range of 0.5–0.8 Mya during the Mid-Pleistocene transition period (0.60–1.25 Mya, Figure 3). This period is considered a crucial turning point in global climate [29]. During this time, there was a shift in the periodicity of Earth’s glacial-interglacial cycles, changing from 41,000 years to nearly 100,000, years resulting in more intense and significant glaciations and sea-level changes [59], which resulted in ongoing changes to the habitats of marine organisms. 

We calculated the substitution rates of some mitochondrial and nuclear genes/sequences using the calibration based on the final closure of the Isthmus of Panama in this study. This rate of *COI* is 6.2 × 10^−3^, slightly larger than that of squat lobsters (genus *Leiogalathea*) which is 5.6 × 10^−3^ [56]. The substitution rate of *16S* is 1.9 × 10^−3^, which is identical to that of the *Leiogalathea* and *Paramunida* squat lobsters [56,60]. The substitute rates of *18S* and *H3* are roughly close to those of *Leiogalathea*, but considering the low nucleotide variations among these sister species, more species should be involved to estimate the substitution rates of nuclear genes in Porcellanidae.

### 4.3. Demographic History and Gene Flow

Tajima’s D test and Fu’s Fs analysis were employed for neutral evolution, and BSP analyses were increasingly used to reconstruct historical demographic expansions. In this study, the results of Fu’s Fs and Tajima’s D tests did not support demographic expansion. However, the BSP analyses of all populations indicated that *P. haswelli* might follow the post-LGM (the Last Glacial Maximum) expansion pattern, as the effective population size of this species increased at approximately 12.5 Kya. When comparing the expansion times with the Pleistocene glacial-interglacial cycles, it was observed that the estimated time for two shrimp species (*Feneropenaeus chinensis* and *Exopalaemon carinicauda*) aligns closely with the onset of the Last Glacial Maximum (LGM) approximately 20,000 years ago [18,61,62]. This suggests a strong influence of the most recent glacial maximum (LGM) on these populations. During glacial periods, populations were likely confined to glacial refugia within a contracted habitat, resulting in a reduction in genetic diversity [63,64]. 

Regarding the results of gene flow Nm studies, genetic drift has been considered to be the main factor driving genetic differentiation between populations when Nm < 1, while Nm > 4 suggests that there is strong gene flow between populations that plays a homogenising role and to some extent impedes the genetic differentiation between populations [65,66]. The dispersal of species is also influenced by the dynamics of marine currents. The North Western Pacific (NWP) region exhibits intricate oceanic current systems along its continental margins, and seasonal circulation patterns are primarily driven by monsoon winds. This leads to the exchange of water between the East China Sea and the South China Sea via the Taiwan Strait. During the summer season, the Taiwan Warm Current and the Yellow Sea Warm Current facilitate the northward transport of planktonic larvae from the South China Sea to the East China Sea [67]. Conversely, during the winter monsoon period, coastal waters flowing southward through the Taiwan Strait promote extensive gene flow among populations along the China coastal seas [68,69,70]. The seasonal variation in zooplankton species composition and abundance in the marine environment is well documented [71,72]. Gong et al. (2015) found that, during the warm water season (April–September), there is a higher diversity and abundance of zooplankton larvae, including zoaea in Zhanjiang Bay [73]. The seasonal variation of biomass of porcelain crabs larvae may result in asymmetric gene flow on both sides of the Taiwan Strait. The Nm from south to north is eight times higher than the gene flow from north to south. In this study, gene flow from the Beibu Bay population to other geographical populations was very small and may be related to ocean currents. Beibu Bay is a small and relatively isolated sea area, being connected with coastal waters of northern South China Sea through the narrow Qiongzhou Strait. The genetic connectivity of invertebrate fauna between Beibu Bay and other sea areas has long been debated, while some population genetic studies suggested that the Qiongzhou Strait might act as a geographic barrier [9,74], although the species composition of some special faunas might be similar on both sides of the strait. Our study indicates that the bidirectional gene flow between Beibu Bay and the northern South China Sea is asymmetrical, and the Nm from the northern South China Sea to the Beibu Bay is 39–67 times higher than that from the reverse direction. Throughout the year, the ocean current in the Qiongzhou Strait essentially travels east to west, and the larvae originating in Beibu Bay cannot freely disperse into the eastern South China Sea due to the influence of counter-clockwise circulation patterns [20,75]. Similar genetic differentiation has been observed in other marine invertebrates inhabiting Beibu Bay, such as *Crassostrea ariakensis* [76], *Marsupenaeus japonicus* [11], *Scylla paramamosain* [9] and *Penaeus monodon* [74], when compared to populations in other localities.

## 5. Materials and Methods

### 5.1. Collections and Morphological Examination

Specimens were collected along the coast of southern China mainland and Hainan Island from the rocky intertidal and from coral reefs (Figure 1). A total of 135 individuals were collected from the survey of marine biodiversity in 2011–2021. All samples were preserved in >80% ethanol after collection, and were deposited in the Marine Biological Museum, Chinese Academy of Sciences, Qingdao, China. An intertidal species *P. polychaetus* Dong, Li and Osawa, 2010 [77] from Hainan, which is morphologically close to the LH complex, was involved for a comparative study. Sample details are summarised in Table 1.

The identification of *P. lamarckii* and *P. haswelli* follows the descriptions by Kropp 1984, Haig 1992, Osawa and Chan 2010 [25,27,28]. The terminology used for morphological descriptions mainly followed Osawa (2007) and Osawa and Chan (2010) [23,25]. The size of the specimens is given as carapace length, which was measured along the dorsal midline from the tip of the rostrum to the posterior margin of the carapace.

### 5.2. DNA Extraction, Sequencing and Molecular Data 

Abdominal muscle or walking legs samples were collected and stored individually in 95% ethanol for further analysis. Total DNA was extracted using the EasyPure Marine Animal Genomic DNA Kit (TransGen) following the manufacturer’s instructions. A fragment of the ribosomal *16S rDNA* was amplified using primers 16Sar-5′ (CGCCTGTTTATCAAAAACAT) and 16Sbr-5′ (CCGGTCTGAACTCAGATCACGT) [78]. A fragment of *cytochrome oxidase I* (*COI*) was amplified using primers HCO2198-5′ (TAAACTTCAGGGTGACCAAAAAATCA) [79] and M1321-5′ (TRRNGAYGAYCARRTTTATAATGT) (designed by Kathleen Saint). A fragment of *cytochrome b* (*Cytb*) was amplified using primers UCYTB151F-5′ (TGTGGRGCNACYGTWATYACTAA) and UCYTB270R-5′ (AANAGGAARTAYCAYTCNGGYTG) [80]. A fragment of the ribosomal *18S rDNA* was amplified using primers 18Sa0.79-5′ (TTAGAGTGCTYAAAGC) and 18Sbi-5′ (AGTCTCGTTCGTTATCGGA) [81]. A fragment of *Histone 3* (*H3*) was amplified using primers H3F-5′ (ATGGCTCGTACCAAGCAGACVGC) and H3R-5′ (ATATCCTTRGGCATRATRGTGAC) [82]. Polymerase chain reaction (PCR) amplification was carried out in a reaction mix containing 1.2 µL of template DNA, 12.5 µL of *ApexHF* HS DNA Polymerase FS Master Mix, 1 µL of each primer (10 mM) and sterile distilled H2O to a total volume of 25 µL. The thermal profile was as follows: initial denaturation for 5 min at 98 °C, followed by 40 cycles of denaturation at 98 °C for 30 s, annealing at 50 °C for 40 s, extension at 72 °C for 40 s, and a final extension at 72 °C for 2 min. PCR multiple products were purified and then sequenced on an ABI 3730 × l DNA analyser by Beijing Tsingke Biotech Co., Ltd., Beijing, China. The obtained haplotype sequences were submitted to GenBank (Appendix A).

The obtained original sequences were checked and revised in DNAStar software [83]. We additionally downloaded the complete mitochondrial genome (Accession numbers NC025572) of *P. haswelli* from Australia deposited in GenBank [84]. Then, the MEGA v7 Sequence Alignment Editor [85] was used to trim primers and excess regions from DNA sequences. Three protein-coding DNA regions (*COI*, *Cytb* and *H3*) were aligned and trimmed with ClustalW [86]. The ribosomal fragments (*16S* and *18S*) were aligned and trimmed with Mafft v7 [87] using the profile alignment method. 

The final alignment length was 547 base pairs (bp) for *COI*, 377 bp for *Cytb*, 441 bp for *16S*, 402 bp for *18S* and 315 bp for *H3*.

The three mitochondrial genes were concatenated and aligned to generate the mito dataset, and a second dataset consisted of these mitochondrial genes and the two nuclear ones to generate all dataset by PhyloSuite v1.2.2 [88] (Appendix A). The two concatenated datasets were used for subsequent partitioned population genetic and phylogenetic analysis.

### 5.3. Population Genetics

Species-specific population genetic diversity, including the number of haplotypes (h), haplotype diversity (Hd) and nucleotide diversity (π), were estimated for each gene sequence using DnaSP v6.11.01 [89].

The relationship between the LH complex and *P. polychaetus* was then recovered using a principal component analysis (PCA) with alldataset. The PCA was implemented through the R package “adegenet” v2.0.1 [90] and the function fasta2genlight extracts SNPs from Mito dataset with FASTA format in the software R v4.2.2 [91].

To visualise the genealogical relationships among haplotypes as well as the lineages recovered by phylogenetic analysis (see Section 5.4) in terms of geographic distribution, the haplotype network of genetic connections between the geographic groups and species was generated with the TCS method in PopART v1.7 [92] using *COI* sequences. 

The genetic distance within and among populations of all *Petrolisthes* species were obtained using the Kimura two-parameter model in MEGA v7 using *COI* sequences. The population pairwise F_ST_ was estimated and tested for significance with 10,000 permutations. For *P. haswelli*, Tajima’s D [93] and Fu’s Fs [94] were applied to test the selective neutrality of genetic markers in Arlequin v3.11 [95] with 10,000 simulations. Additionally, these estimators are sensitive to recent population expansion or bottlenecks. The interim trends in effective population size were reconstructed with Bayesian Skyride Plot [96] implemented in BEAST v2.6.3 [97] using 10,000,000 Markov chain Monte Carlo (MCMC) steps, sampled every 1000 generations. We used substitution rates of 0.5% per million years of the Mito dataset specified in BEAST v2.6.3, which was estimated in the divergence time analysis (see Section 5.5).

### 5.4. Phylogenetic Analyses and Species Delimitation

Phylogenetic analyses were conducted with all dataset using Bayesian (BI) and maximum likelihood (ML) methods. In addition, we constructed phylogenetic trees using each of the five gene datasets separately. Additionally, we obtained a phylogeny based on the Mito dataset for comparison. The best partitioning schemes and best-fit substitution models for each partition in all dataset were identified using ModelFinder [98] and PartitionFinder 2 [99] based on the Bayesian information criterion [100]. All the analyses mentioned above were performed using the integrated and scalable desktop platform PhyloSuites v1.2.2. ML trees were inferred in IQtrees v1.6.12 [101], with 1000 bootstrap (BS) replicates to estimate the BS values of nodes. Bayesian inference phylogenies were inferred using MrBayes 3.2.6 [102]. Two independent runs were executed with four Markov chain Monte Carlo (MCMC) for 1,000,000 generations, sampling every 100 generations, in which the initial 25% of sampled data were discarded as burn-in. *Pisdia serratifrons* (Stimpson, 1858) [31] (sequences were obtained in this study, see Appendix A) and *Pisidia magdalenensis* (Glassell, 1936) [103] (sequences were downloaded from the GenBank, see Appendix A) were used as outgroups. The phylogenetic trees were visualised with FigTree v1.4.2 [104] and beautified by iTOL webserver [105]. 

Species boundaries were first estimated on the basis of genetic distance of *COI* sequences among different lineages recovered above. Secondly, we used the general mixed Yule coalescent (GMYC) method [106] and a Bayesian implementation of the Poisson tree processes model (bPTP) [107]. Both analyses were implemented in the GMYC and bPTP web servers (http://species.h-its.org/, accessed on 9 August 2023). For the GMYC analysis, an ultrametric tree based on all dataset generated by BEAST v2.6.3 was used as input. The single-threshold model was used in GYMC method. In order to run bPTP analysis, a Bayesian inference (BI) and a maximum likelihood (ML) tree based on all dataset were used as inputs for comparison. The number of MCMC generations was 100,000, the thinning was set to 100, and burn-in was set to 25% of the initial samples. The convergence of the parameters was checked after the run.

### 5.5. Divergence Time Estimation

The dates of the divergence between the major clades were estimated in BEAST v2.6.3 with all dataset. This program uses relaxed clock models and allows for missing data and the flexibility of model parameters. We implemented an uncorrelated relaxed log normal clock model, with values sampled from a distribution having a mean of 0.002 and a standard deviation of 0.1, for the 16S clock rate. These rates were derived from previously reported mitochondrial DNA substitution rates for Galatheoidea [56,60] and Porcellanids are closely related to these Anomuran Decapods. Mean substitution rates were estimated for each gene. The germinate populations of a congener *Petrolisthes armatus* (Gibbes, 1850) [108] were used to obtain a rate of divergence for each DNA region. *COI*, *16S*, *Cytb*, *18S* and *H3* sequences were downloaded from the GenBank (see Appendix A) [8,13]. *Petrolisthes armatus* of a congener of the *P.lamarckii–P*. *haswelli* complex shows the least transisthmian divergence [13,109] which is the least divergence within Porcellanidae and is thus likely to have been separated during the final stages of Isthmus of Panama completion, 3 Ma [110,111]. Nevertheless, Montes et al. (2015) suggested that only narrow water passages between the eastern Pacific and the Caribbean Sea after 12.5 Ma [112]. Hiller and Lessios (2020) suggested that their offset was assigned an exponential prior with a range from 3 to 12.5 Ma in a lognormal uncorrelated relaxed clock [8]. Here, we used the same parameter setting as Hiller and Lessios (2020) [8]. Tracer v1.7.1 [113] was used to verify the effective sample size (ESS) values of all sampled parameters. After the first 25% of trees were discarded as burn-in, the maximum clade credibility tree with median nodal height was generated using TreeAnnotator 2.6.3 [97]. The time-scaled tree was mapped against geological time and visualised using the R packages strap [114] and phytools [115] in the software R v4.2.2.

### 5.6. Demographic History and Gene Flow of Petrolisthes haswelli

According to the morphological and molecular analysis, *P. haswelli* is the most widely distributed species along the subtropical Chinese coast and exhibits different colour patterns (see Figure 7B–E). Therefore, we are interested in their genetic connectivity among individuals in different geographic areas. We first grouped the populations of this species into five area-populations including XG (including XG region), DS (including DS region), EC (including HZ and MW regions), BH (including FCG, WZD, XW and DZ regions), HN (including JJD and LHT regions). A coalescent-based method (Migrate-n) was then used to estimate the migration. It better estimates the recent migration affected by recent changes [116]. Analyses estimating migration rates were implemented in Migrate-n 3.6.11 [117] based on the Mito dataset. Migrate-n analyses were conducted using Bayesian inference with slice sampling. The following parameters were used for Markov chain Monte Carlo runs: Brownian motion with constant mutation rate, recording 500,000 steps at 20-step increments (total steps 1,000,000) and the first 10,000 genealogies were discarded as burn in and running 3 replicates. We used the static heating with four temperatures as specified by the command in the program for even spacing (temperatures were 1.00, 1.5, 3.0, and 1,000,000.0). Posterior distribution, autocorrelation and effective sample size (ESS) were checked for convergence. Description of parameters and commands are described in the Migrate-n program manual [118].

## Figures and Tables

**Figure 1 ijms-24-15843-f001:**
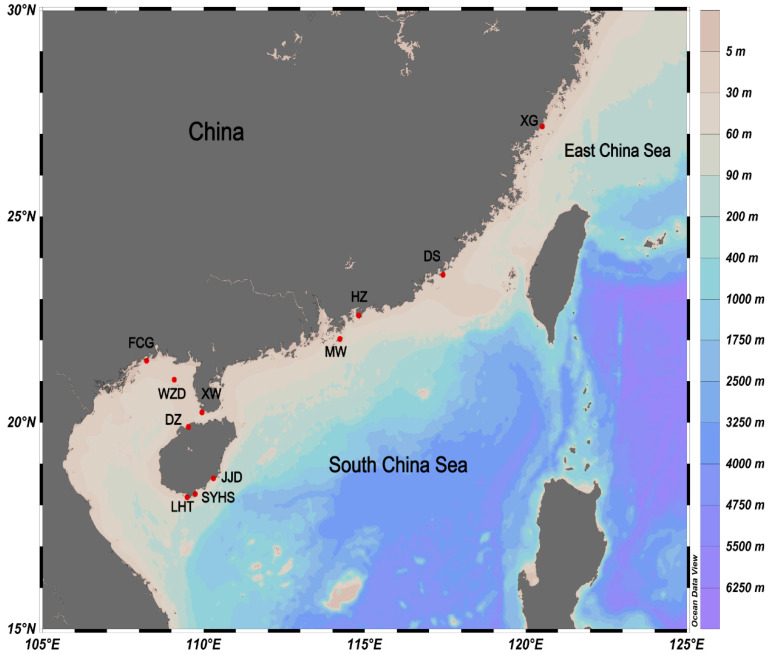
The sampling locations of four *Petrolisthes* species (*Petrolisthes haswelli*, *P. lamarckii*, *P. polychaetus* and *P. shanyingi* sp. nov.) in this study.

**Figure 2 ijms-24-15843-f002:**
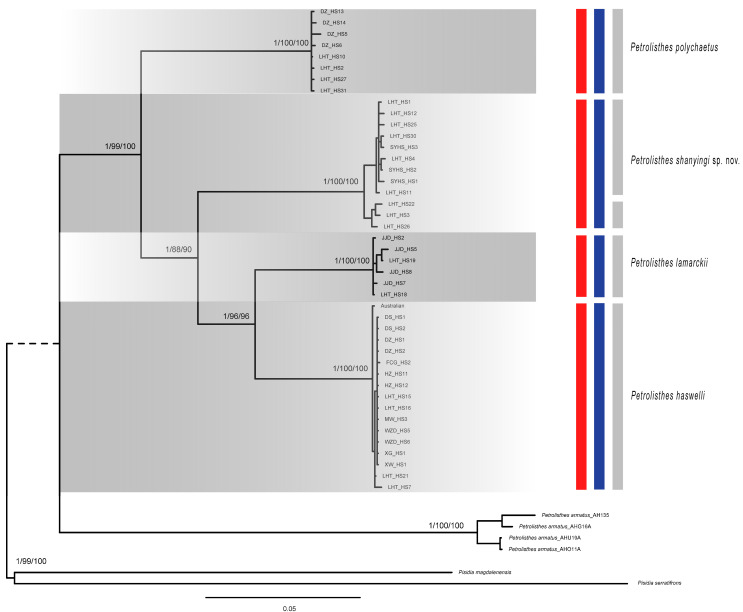
Phylogenetic tree based on concatenated mitochondrial (*16S*, *COI* and *Cytb*) and nuclear (*18S* and *H3*) sequences of *Petrolisthes*. Numbers above branches indicate posterior probability, SH-aLRT support (%) and ultrafast bootstrap support (%) values, respectively. The results of the species delimitation analyses are also indicated by the coloured bars: bPTP with ML tree in red, bPTP with Bayesian tree in blue and GMYC in grey. Branch tips are labelled according to the morphospecies. *Pisdia* was used as outgroup.

**Figure 3 ijms-24-15843-f003:**
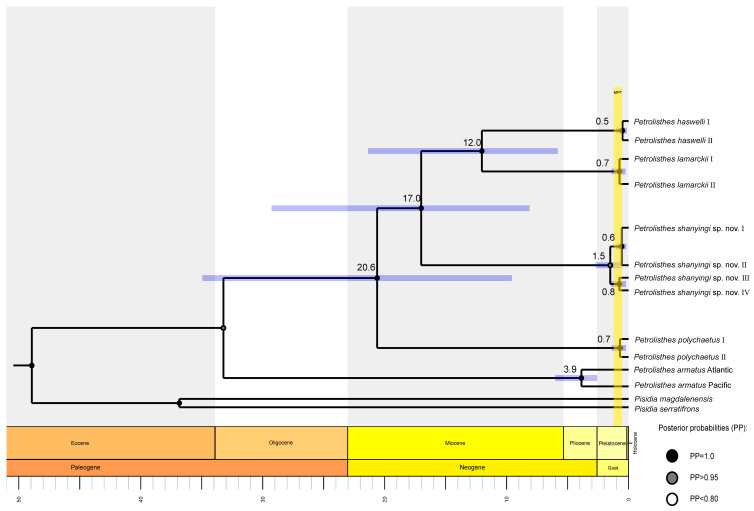
Time-scaled tree inferred by BEAST, showing 95% HPD credibility intervals for each well-supported recovered node by purple horizontal lines. Numbers above branches represent the average lineage divergence times; posterior probabilities (PP) are annotated using different coloured dots. The interval of the Mid-Pleistocene transition (MPT) is plotted based on Petrick et al. (2019) [29].

**Figure 4 ijms-24-15843-f004:**
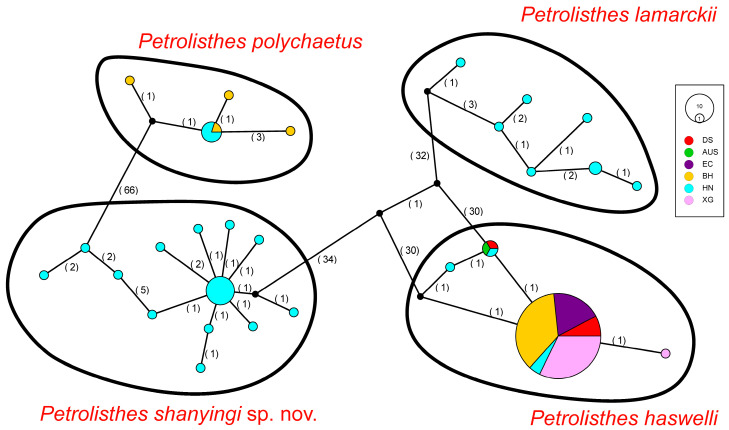
Haplotype network of *Petrolisthes haswelli, P. lamarckii, P. polychaetus* and *P. shanyingi* sp. nov., based on the *COI* dataset. Each circle represents a unique haplotype and its size reflects the number of individuals carrying that haplotype. Colours indicate locality according to the legend and small black circles represent hypothetical ancestors or unsampled haplotypes. Segment sizes within circles indicate the distribution of haplotypes among different regions. Numbers between circles represent the number of mutational steps. The haplotypes of each species are depicted in black circumferences.

**Figure 5 ijms-24-15843-f005:**
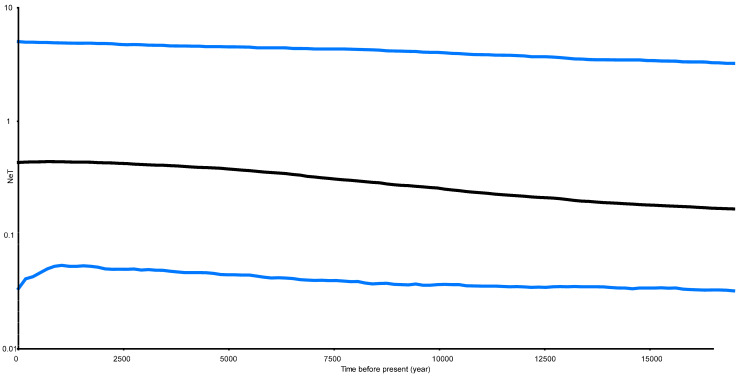
Bayesian skyline plot of the effective population sizes over time for *Petrolisthes haswelli.* The X axis is the time scale in units of years, and the Y axis is the estimated effective population size in units of NeT, the product of the effective population size and the generation length in years (log transformed). Black lines represent median estimates while the blue lines show the 95% highest posterior density (HPD) limits.

**Figure 6 ijms-24-15843-f006:**
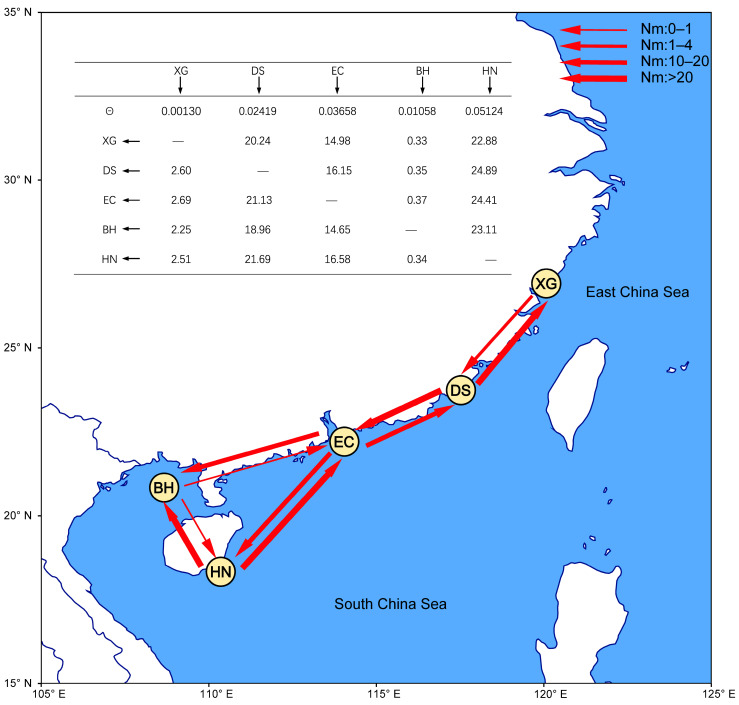
The diagram depicts gene flow between populations based on Migrate-n. *θ* represents the parameters of the mutation-scaled effective population size and the black arrow represents the direction of gene flow. The red arrows indicate the direction of gene flow, and the magnitude of gene flow is represented by different types of arrows as shown in the legend.

**Figure 7 ijms-24-15843-f007:**
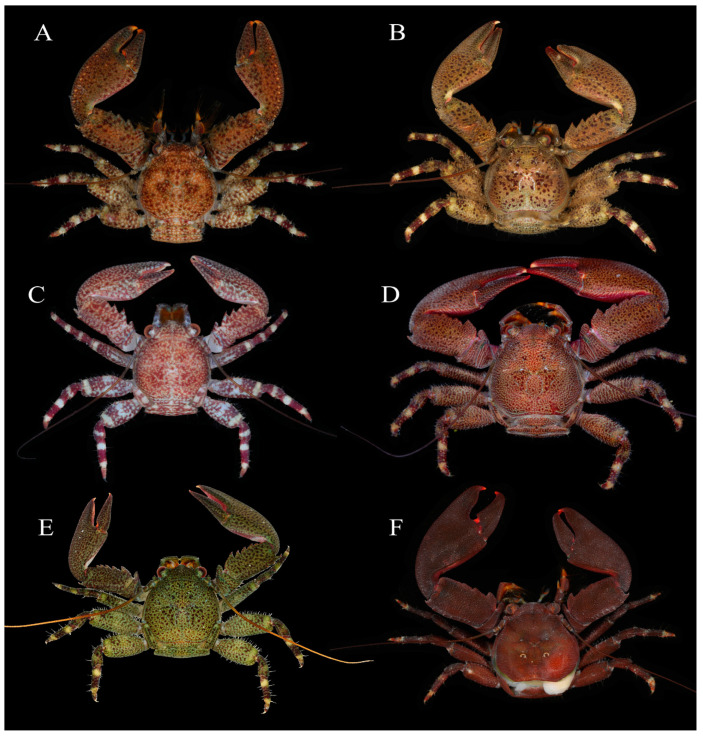
Specimens in fresh colour, dorsal view. (**A**), *Petrolisthes lamarckii*, Jiajing Island, Hainan Province. (**B**), *Petrolisthes haswelli*, Danzhou, Hainan Province. (**C**), *Petrolisthes haswelli*, juvenile, Danzhou, Hainan Province. (**D**), *Petrolisthes haswelli*, Dongshan, Fujian Province. (**E**), *Petrolisthes haswelli*, Weizhou Island, Guangxi Province. (**F**), *Petrolisthes shanyingi* sp. nov., paratype, Sanya, Hainan Province.

**Figure 8 ijms-24-15843-f008:**
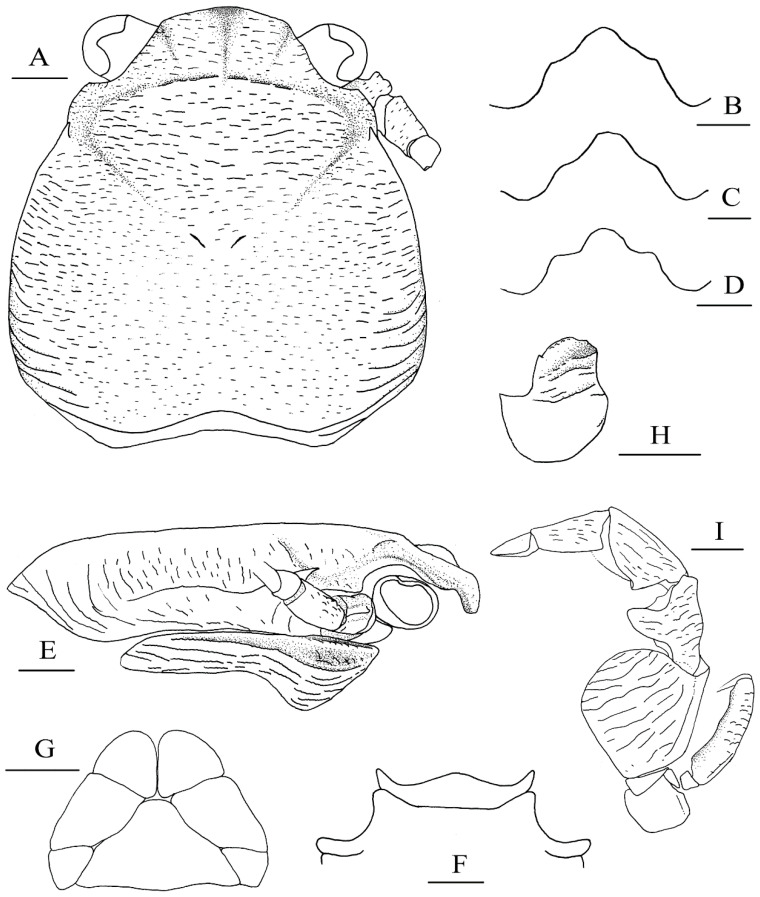
*Petrolisthes shanyingi* sp. nov., holotype, LHT-HS5, male (**A**,**B**,**E**–**I**); *Petrolisthes lamarckii*, JJD-HS8, male (**C**); *Petrolisthes polychaetus*, LHT-HS27, male (**D**). (**A**), carapace and right antennal peduncle, dorsal view; (**B**–**D**), rostrum, anterodorsal view; (**E**), right carapace and pterygostomian, lateral view; (**F**), third and fourth thoracic sternite, ventral view; (**G**), telson, external view; (**H**), basal article of left antennular peduncle, ventral view; (**I**), left third maxilliped, ventral view. Scales equal 1.0 mm.

**Figure 9 ijms-24-15843-f009:**
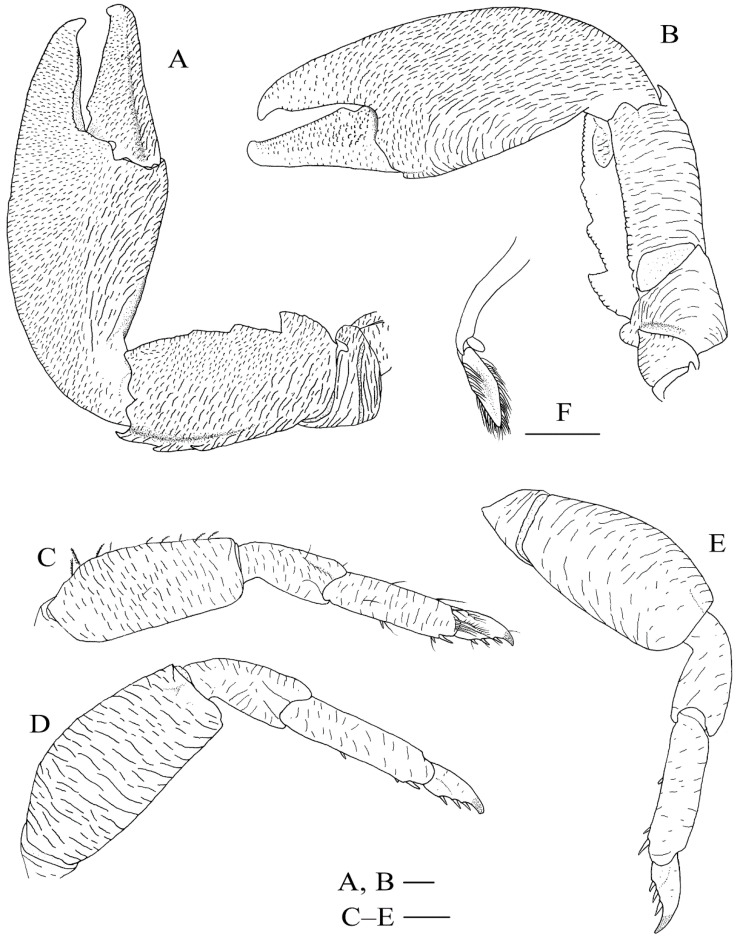
*Petrolisthes shanyingi* sp. nov., holotype, LHT-HS5, male. (**A**) left cheliped, dorsal view; (**B**), left cheliped, ventral view; (**C**), right first ambulatory leg (with setae), lateral view; (**D**), right second ambulatory leg, lateral view; (**E**), right third ambulatory leg, lateral view; (**F**), left pleopod on second abdominal segment, dorsal view. Scales equal 1.0 mm.

**Table 1 ijms-24-15843-t001:** List of localities inhabited by *Petrolisthes haswelli, P. lamarckii, P. polychaetus* and *P. shanyingi* sp. nov. (name, abbreviation and geographical position) and genetic characterisation of a sequenced individual.

Locality	S	Latitude, Longitude	Taxon	*COI*				*16S*				*CYTB*				*18S*				*H3*			
				N_1_	H_1_	Hd_1_	π_1_	N_2_	H_2_	Hd_2_	π_2_	N_3_	H_3_	Hd_3_	π_3_	N_4_	H_4_	Hd_4_	π_4_	N_5_	H_5_	Hd_5_	π_5_
Xiaguan (XG)	33	27.19° N 120.51° E	*P. haswelli*	30	2	0.067	0.00012	22	1	0	0	33	1	0	0	2	1	0	0	21	1	0	0



Dongshan (DS)	8	23.59° N 117.43° E	*P. haswelli*	8	2	0.250	0.00046	8	2	0.250	0.00057	8	1	0	0	7	1	0	0	8	1	0	0



Huizhou (HZ)	11	22.60° N 114.81° E	*P. haswelli*	11	1	0	0	11	1	0	0	8	2	0.250	0.00067	2	1	0	0	5	1	0	0



Miaowan (MW)	6	22.03° N 114.23° E	*P. haswelli*	6	1	0	0	6	1	0	0	6	1	0	0	2	1	0	0	1	1	0	0



Fangcheng (FCG)	4	21.50° N 108.23° E	*P. haswelli*	4	1	0	0	4	1	0	0	2	2	1.000	0.00265	2	1	0	0	3	1	0	0



Weizhou (WZD)	12	21.04° N 109.09° E	*P. haswelli*	12	1	0	0	3	1	0	0	2	1	0	0	4	1	0	0	3	1	0	0



Xuwen (XW)	7	20.25° N 109.95° E	*P. haswelli*	7	1	0	0	2	1	0	0	2	1	0	0	1	1	0	0	2	1	0	0



Danzhou (DZ)	14	19.90° N 109.53° E	*P. haswelli*	10	1	0	0	6	1	0	0	2	1	0	0	2	1	0	0	5	1	0	0
			*P. polychaetus*	4	4	1.000	0.00548	4	2	0.500	0.00113	4	4	1.000	0.00663	4	1	0	0	4	1	0	0


Jiajingdao (JJD)	6	18.65° N 110.30° E	*P. haswelli*	1	1	0	0																
			*P. lamarckii*	5	5	1.000	0.00841	5	5	1.000	0.00363	4	3	0.833	0.00531	5	1	0	0	5	1	0	0


Houhai (SYHS)	3	18.27° N 109.74° E	*P. shanyingi* sp. nov.	3	3	1.000	0.00366	3	1	0	0	3	3	1.000	0.00884	3	1	0	0	3	1	0	0



Luhuitou (LHT)	31	18.19° N 109.49° E	*P. haswelli*	5	3	0.700	0.00184	5	3	0.700	0.00272	5	2	0.400	0.00106	4	1	0	0	5	1	0	0
			*P. lamarckii*	3	3	1.000	0.00366	3	1	0	0	2	2	1.000	0.00531	3	1	0	0	3	1	0	0
			*P. polychaetus*	4	1	0	0	4	1	0	0	4	4	1.000	0.00398	4	1	0	0	4	1	0	0
			*P. shanyingi* sp. nov.	19	11	0.789	0.00552	19	8	0.673	0.00223	13	11	0.962	0.01051	12	1	0	0	19	1	0	0
Total	135		*P. haswelli*	94	4	0.084	0.00019	67	3	0.088	0.00027	68	4	0.087	0.00023	26	1	0	0	53	1	0	0
			*P. lamarckii*	8	7	0.964	0.00620	8	5	0.786	0.00227	6	4	0.800	0.00460	8	1	0	0	8	1	0	0
			*P. polychaetus*	8	4	0.643	0.00274	8	2	0.250	0.00057	8	7	0.964	0.00531	8	1	0	0	8	1	0	0
			*P. shanyingi* sp. nov.	22	13	0.805	0.00530	22	8	0.602	0.00194	16	14	0.975	0.01017	15	1	0	0	22	1	0	0

S is the number of porcelain crabs obtained at sampling points. Nx is the number of individuals used for sequencing at different DNA markers, Hx is the number of haplotypes, Hdx is the haplotype diversity and πx is the nucleotide diversity. (x stands for different gene markers and its values are 1, 2, 3, 4, and 5.

**Table 2 ijms-24-15843-t002:** Matrix of pairwise F_ST_ (below diagonal) and pairwise genetic distance (above the diagonal) values among different populations. (* *p* < 0.05).

	XG	DS	EC	BH	HN
XG		0.000	0.000	0.000	0.001
DS	0.042		0.000	0.000	0.001
EC	−0.021	0.102		0.000	0.001
BH	0.003	0.213	0.000		0.001
HN	0.288	−0.057	0.320	0.482 *	

## Data Availability

The datasets generated for this study can be found in the GenBank (http://www.ncbi.nlm.nih.gov/genbank).

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
