# Peer review of "Phylogenetics and Population Genetics of the Petrolisthes lamarckii–P. haswelli Complex in China: Old Lineage and New Species"

_ijms, 2023, doi:10.3390/ijms242115843_

Round 1

Reviewer 1 Report

Comments and Suggestions for Authors

The present study is a good work using integrative methods combining the taxonomic, phylogenetic and population genetic analyses to explore the biodiversity and distribution pattern of the species complex of porcelain crabs, Petrolisthes lamarckii-P. haswelli, on the coasts of China. It is well organized in general. Specific suggestions and comments are shown below. 

Line 11.  “a pair of sister species”, replaced by “a pair of sister species of porcelain crabs”.

Line 18.  “from all species”, replaced by “from all the three species”.

Line 25.  “12.5 thousand years ago (kya)”, replaced by “12.5 thousand years ago (Kya)”??

Line 34.  “this marginal sea provides”, replaced by “these marginal seas provide”.

Line 70.  “its large body”, replaced by “the relatively large body”.

Lines 78-80.  Please give the literature indicating the sentence “The two species, especially P. haswelli, can form large populations in the intertidal rocky habitats, thus playing an important part in the subtropical coastal ecosystems”. 

Line 101.  “describe cryptic species”, replaced by “describe a cryptic new species”.

Line 123. “size of the carapace”, replaced by “size of the specimens”.

Line 144. What does “Tsingke” means??

Lines 197-198.  Are the describers needed for Pisdia serratifrons (Stimpson, 1858) and Pisidia magdalenensis (Glassell, 1936)?? Please unify in the entire text if the describers are given for all species names or not.

Line 258.  What are “four Petrolisthes species”?? If they include a new species described in this species, show all of the species names clearly.

Line 268.  “Each species only has”, replaced by “Each species has only”.

Line 286.  “(Figure S1-S6)..”, replaced by “(Figure S1-S6).”.

Line 290.  “a closely relationship among species”, replaced by “a closer relationship”.

Line 300. “recovered five and four Petrolisthes molecular lineages”, replaced by “recovered four or five Petrolisthes molecular lineages”??

Line 359.  “The Nm from”, replaced by “the Nm from”.

Line 372.  “(Leach, 1820)”, do not be italicized.

Line 407.  “numerous short rugae”, replaced by “numerous weak, short rugae”??

Line 408.  “3 to 5 teeth on dorsoflexor margin”, replaced by “3 to 5 (usually 4) teeth on dorsoflexor margin”??

Lines 410-412.  “lateral flexor margin with acute distal spine on first and second legs; carpi armed with distal spine on extensor margin of first leg;”, replaced by “lateral flexor margin with acute distal spine on first and second legs, but unarmed on third leg; carpi armed with distal spine on extensor margin of only first leg;”??

Line 415.  “(orange based on our specimens)”, replaced by “(light brown based on our specimens)”??

Line 416.  “row of color spots”, replaced by “row of orange spots”??

Line 426.  “the merus of the second ambulatory leg”, replaced by “the merus of the first ambulatory leg”.

Lines 428-429.  “the merus of the second ambulatory leg”, replaced by “the merus of the first ambulatory leg”.

Line 430.  “row of color spots”, replaced by “row of orange spots”??

Line 436.  “deposited”, replaced by “was deposited”.

Line 437.  “type material”, replaced by “the type material”.

Line 438.  “the type material of the P. lamarckii”, replaced by “the holotype of P. lamarckii”. Additionally, does the type material of P. lamarckii only include the holotype??

Line 440.  “the western and northern coast”, replaced by “the western or northern coast”.

Line 441.  “in a poor condition”, replaced by “now in poor condition”.

Line 442.  “were left”, replaced by “are left”.

Line 443.  “no spine”, replaced by “no spines”.

Lines 445-447.  The holotype of P. lamarckii exists at present, and thus a neotype for that taxon is not be needed and do not never become the type material.

Lines 447 and 454.  “Kropp’s (1986)”, replaced by “Kropp’s (1984)”.

Lines 449-451.  Kropp (1984) described Petrolisthes lamarkii as having 3 low teeth and infrequently a four distal tooth on the anterior margin of the cheliped carpus.  Additionally, based on the Marianas specimens, the cheliped palm has a row of irregularly spaced light blue spots along the dorso-anterior margin. These may indicate that the specimens of “P. lamarckii” Kropp (1984) examined include more than one species exclusive of P. lamarkii sensu stricto and other cryptic undescribed species, as well as the present new species if Kropp’s (1984) material has variations in fresh coloration.

Lines 453-454.  Leach (1820) might overlook the distal fourth tooth on the anterior margin of the cheliped carpus in the type material (only the holotype??) because the tooth is occasionally very low.  However, this is only a speculation.

Lines 464-467.  The morphological distinctions between P. lamarckii and P. haswelli mentioned here have been already discussed in Osawa & Chan (2010). Please put this literature as a reference here. 

Line 500.  “two parts”, replaced by “2 parts”.

Line 501.  “dorsal half of lateral surface depressed”, replaced by “dorsal half of surface depressed”.

Line 503.  “trilobite”, replaced by “trilobate”.

Line 503.  “seven plates”, replaced by “7 plates”.

Line 515.  “ventral surface”, replaced by “lateral surface”.

Lines 517-518.  “ventral surface with rugae along extensor margin. Propodus and dactyl relative smooth on ventral surfaces”, replaced by “lateral surface with rugae entirely. Propodus with sparse short rugae on lateral surface. Dactlylus relatively smooth on lateral surface”.

Line 533.  “forming shallow submarginal groove”, replaced by “with shallow submarginal groove”.

Line 543.  “P2 carpus and P4 carpus”, replaced by “P2 and P4 carpi”.

Line 553.  “Orange, brown red”, replaced by “Brown, reddish brown”??

Lines 563-568.  The morphological characters shared by the three species need to include the carapace and chelipeds covered with weak rugae (striae) on the dorsal surfaces.

Line 568.  “is previously”, replaced by “was previously”.

Line 573.  “may acute”, replaced by “may be acute”.

Lines 573-574.  “the P2 merus without or only has small blunt distal spine”, replaced by “the P2 merus is unarmed or only has a small blunt distal spine.  Please also unify the terms of ambulatory legs entirely on the text, “first to third ambulatory legs” or “P2-4 (second to fourth pereopods)”.

Lines 577-578.  “the P2 merus bear strong distal spine on the lateral flexor margin; and the P2 carpus is armed with acute distal spine on the extensor margin”, replaced by “the P2 merus bears a more developed distal spine on the lateral flexor margin; and the P2 carpus is armed with an acute distal spine on the extensor margin”.

Line 582.  “flexor margin”, replaced by “dorsoflexor margin”.

Line 583.  “is unarmed on the P2 carpus”, replaced by “no disto-extensor spines on the P2 carpus”.

Line 587.  “more produced”, replaced by “more strongly produced anteriorly”.

Line 589.  “both rounded corner”, replaced by “both a rounded corner”.

Line 590.  Is the spine on the lateral distoflexor margins of the P2 and P3 meri is smaller or less developed in the new species than in P. lamarckii and P. haswelli??

Line 591.  “no such spines. Comparatively, the distal spine”, replaced by “no such spine, whereas the distal spine”.

Line 593.  “but according to”, replaced by “but judging from”.

Line 614. “microscopic”, replaced by “infinitestimal”??

Line 627.  “the Petrolisthes”, replaced by “Petrolisthes species”.

Lines 630-633.  Is Petrolisthes armatus a species inhabiting the tropical environments like the present three species, although it is primary distributed in the western Atlantic??

Line 637.  “Munidid and Munidopsid”, replaced by “munidid and munidopsid”.

Lines 647-649.  It is reasonable that P. lamarckii (one of two morphological forms by Osawa & Chan, 2010) is more closely related to P. hawselli than P. shanyingi sp. nov. because P. lamarckii and P. haswelli are very similar to each other in the nearly entire morphology.

Line 663.  “The deep-sea squat lobster was”, replaced by “The deep-sea squat lobsters were”.

Lines 673-674.  I could not understand the phrase “which may have had a profound impact on the evolution of biota evolution”. Please consider to modify this. 

Lines 678-679.  “identical with the Leiogalathea and Paramunida squat lobsters”, replaced by “which is identical with that of Leiogalathea and Paramunida squat lobsters”.

Line 689.  “12.5 kya”, replaced by “12.5 Kya”??

Line 729.  “such as the”, replaced by “such as”.

Line 731.  “the Penaeus monodon”, replaced by “Penaeus monodon”.

Comments on the Quality of English Language

Please see the suggestions mentioned above.

Reviewer 2 Report

Comments and Suggestions for Authors

This manuscript deals with the systematics and taxonomy of the Petrolisthes lamarckii complex, which comprises cryptic species of porcellanid crabs, some of which are widely distributed in the Indo-West Pacific. The molecular data is based on three mitochondrial and two nuclear genes. The manuscript also deals with local analyses of the of geographic distribution of genetic variation in P. haswelli, a member of this complex, within the southern China Sea. The authors describe a new species, previously considered to be a variety within the P. lamarckii complex, which is supported as a monophyletic clade in the phylogeny presented.

The manuscript is relatively well written, but with some confusing paragraphs, which need to be edited. Among the many figures the authors depict sampling localities, the potential direction and magnitude of gene flow in P. haswelli within the southern China Sea, and the effective population size of this species over time. The authors also depict dated phylogenies of three species of the P. lamarckii complex (including the new species), and an additional species, which does not belong to the complex (P polychaetus). A haplotype network showing haplotype distribution of the four species within the sampled region is depicted as well, and images of freshly collected material of the four species used in the phylogenetic analyses were included.

Even though the manuscript is worth publishing, I recommend major revision for the following reasons:

The title and conclusions should be put into context with the studied region and material. The title suggests a population genetics analysis of the whole complex, which would be very impressive given that this complex contains several nominal species (over 7) and other cryptic ones still considered part of the morphological variation of P. lamarckii. The population genetics analyses only deals with populations of P. haswelli in the southern China Sea. Analyses are reasonably done, but the conclusions on past and present gene flow and effective population size end up walking on thin ice because the data does not support such wide conclusions. Authors should be cautious and admit the limited number of individuals per population and population distribution in the western Pacific. They only include one individual from Australia, whose mitochondrial genome is available at GenBank. This is useful to confirm the identity of the P. haswelli material from China, but not to conclude on so many aspects of the population genetics of this species. The local analyses of the gene flow within the China Sea are interesting and so are the conclusions, but they need to be limited to this region only.

The same goes for the phylogenetic analyses of the complex. The authors are missing important species comprising the complex (e.g. fimbriatus, asiaticus), which they did not examine morphologically, but conclude on the status of sister species. P. lamarckii has been documented for decades as a complex of species with wide distribution in the Indo-West Pacific. The inclusion in the phylogeny of P. polychaetus, besides haswelli, lamarckii and the new species, is not well justified. This species does not belong to the complex, as far as I am concerned. The phylogeny should be used to confirm reciprocal monophyly of the new species, haswelli and lamarckii.

A confusing issue is that the authors declare not knowing a priori what species they are dealing with, precisely because they are dealing with a complex, but this would have been solved by looking at a preliminary phylogenetic clustering based on mitochondrial markers. It is not clear at what time they are dealing with 4 species or with the whole group of unidentified individuals, and this makes it difficult to follow up results. Another problem is that the authors talk about most recent common ancestors (MRCA), but they are missing several species of the P. lamarckii complex. Again, status of sister species cannot be concluded unless all (or most) species in a monophyletic clade are included in the phylogeny.

From a taxonomic perspective, while the description of the new species seems to follow all taxonomic mandates, the authors examined a very limited number of specimens. P. lamarckii has been deposited in several musei, including the Paris Museum (MNHN), and the Smithsonian Museum, and I am sure they have been deposited in important Asian musei and institutions like the National Taiwan Ocean University, Keelung, the National Museum of Natural Science, Taichung, the Raffles Museum of Biodiversity Research or the National University of Singapore. More material needs to be examined to consider all intraspecific variation within the P. lamarckii complex.

Specific problems: Some parts lack order, you include Tables 1 and 2 before the map, and under Material and Methods. These correspond to results.

I would leave the description of the new species for a second publication dealing with the labyrinthic taxonomic history of P. lamarckii. More specimens from a large part of the geographic distribution should be examined.

My comments and editions are included in the PDF. They are marked in red colour and comments are directly inked to these markings.

I recommend publication after major revision, leaving the description of the new species for another publication. The molecular data should be justified and treated with caution. It is better to admit some limitations than to go beyond what the data can support (kind of walking on thin ice).  

Comments on the Quality of English Language

Needs some improvement. I did some suggestions. See reviewed PDF

I do not wish to review this manuscript again.

Round 2

Reviewer 2 Report

Comments and Suggestions for Authors

The manuscript did improve but it still needs editorial corrections. 

Authors tried to address most corrections/suggestions, but in some instances they just deleted a word and did not correct. For instance, they deleted the name of the model used to estimate genetic distances (uncorrected p distances). The suggestions pointed at using a better model to estimate distances, like K2P, and not just deleting the name but still mention that genetic distances were calculated, without mentioning how they did it. 

After new minor corrections the manuscript should be ready for publication. 

Comments on the Quality of English Language

Needs minor improvement
